

# Technical Note: revisiting the general calibration of cosmic-ray neutron sensors to estimate soil water content

Maik Heistermann[1], Till Francke[1], Martin Schrön[2], and Sascha E. Oswald[1]

[1]Institute of Environmental Science and Geography, University of Potsdam, Karl-Liebknecht-Straße 24–25, 14476 Potsdam, Germany

[2]UFZ - Helmholtz Centre for Environmental Research GmbH, Dep. Monitoring and Exploration Technologies, Permoserstr. 15, 04318, Leipzig, Germany

**Correspondence:** Maik Heistermann (maik.heistermann@uni-potsdam.de)

**Abstract.** Cosmic-ray neutron sensing (CRNS) is becoming increasingly popular for monitoring soil water content (SWC). To retrieve SWC from observed neutron intensities, local measurements of SWC are typically required to calibrate a location-specific parameter, $N_0$, in the corresponding transfer function. In this study, we develop a generalized conversion function that explicitly takes into account the different factors that govern local neutron intensity. That way, the parameter $N_0$ becomes location-independent, i.e., generally applicable. We demonstrate the feasibility of such a "general calibration function" by analysing 75 CRNS sites from four recently published datasets. Given the choice between the two calibration strategies – local or general – users will wonder which one is preferable. To answer this question, we estimated the resulting uncertainty of the SWC by means of error propagation. While the uncertainty of the local calibration depends on both the local reference SWC itself and its error, the uncertainty of the general calibration is mainly governed by the errors of vegetation biomass and soil bulk density. An interactive online tool is provided to support the decision decide which calibration strategy – local or general – is preferable in the user-specific application context (https://cosmic-sense.github.io/local-or-global).

## 1 Introduction

Cosmic-ray neutron sensing (CRNS) is becoming increasingly popular for monitoring soil water content (SWC). The technology is non-invasive, and has a horizontal footprint of around 150 m together with a vertical penetration depth of typically up to 30 cm. That way, CRNS overcomes a fundamental disadvantage of point measurements - the lack of spatial representativeness.

Yet, the estimation of volumetric soil water content ($\theta$, in m³/m³) from observed neutron count rates ($N$, in counts per hour, cph) requires reference measurements of $\theta$ in order to calibrate the parameter $N_0$ in the functional relationship proposed by Desilets et al. (2010):

$$\theta(N) = \left( \frac{a_0}{\frac{N}{N_0} - a_1} - a_2 \right) \cdot \frac{\varrho_b}{\varrho_w} \tag{1}$$

where $\rho_b$ is the soil bulk density (kg/m³), $\rho_w$ is the density of water (kg/m³), and $a_0$=0.0808 g/g, $a_1$=0.372 and $a_2$=0.115 g/g are constants.





Typically, such reference measurements of $\theta_{\mathrm{cal}}$ consist of 10-20 profiles in the CRNS footprint: within the upper 30 cm of the soil, $\theta$ and $\rho_b$ are measured at different depth increments and within various distances from the neutron detector. In order to obtain a value of $\theta_{\mathrm{cal}}$ that is representative for the detector footprint, the individual measurements at different depths and

25 horizontal distances are averaged by a set of weighting functions (Schrön et al., 2017).

These reference measurements are labour-intensive (roughly one person-day for sampling only, excluding travels and laboratory analysis). More importantly, though, the variability of $\theta$ at different spatial scales makes it difficult to representatively cover a circle of 150 m radius with such a limited number of samples. Any error of $\theta_{\mathrm{cal}}$ is expected to propagate to $\theta(N)$, although systematic studies to that effect do, to our knowledge, not exist, yet.

30 For mobile applications (CRNS roving), e.g. by car (Schrön et al., 2018) or train (Altdorff et al., 2023), the problem becomes even more obvious as obtaining reference measurements along extended roving tracks is practically impossible. Other conditions can make reference measurements difficult to unfeasible, such as access restrictions (e.g. to agricultural fields, private property), or soil properties (stones, roots).

Ideally, the local calibration of $N_0$ could be replaced by a general relationship which takes into account all the factors (apart

35 from SWC) that influence the local neutron intensity and hence any estimate of $N_0$. The key ingredients for such a relationship were already elaborated, in this journal, about eleven years ago. Zreda et al. (2012) outlined a framework for the COSMOS network to account for:

– the dynamic effects of barometric pressure, air humidity and the incoming neutron flux (for which the correction functions are still commonly used, although alternatives were suggested);

40 – the spatial variation of incoming cosmic-ray secondary neutron intensity as governed by the Earth's geomagnetic field and the location in the atmosphere (i.e. altitude), based on a model proposed by Desilets and Zreda (2003);

– the efficiency of the neutron detector, by defining a reference probe (the first COSMOS site in San Pedro) to which all neutron count rates could be scaled.

This concept was further refined by Franz et al. (2013), again in this journal, who suggested a "universal calibration function

45 for determination of soil moisture with cosmic-ray neutrons" to take into account the differences in various hydrogen pools between different sites, namely biomass, soil organic matter (OM) and lattice water (LW). Based on the data from 35 COSMOS sites and 45 calibration dates, Franz et al. (2013) demonstrated the basic feasibility of the concept, although it should be noted that the functional relationship was formally established between neutron intensity and the molar fraction of hydrogen in a support volume (instead of $\theta$). The authors also suggested a detector-specific calibration parameter, $N_s$, which represents the

50 neutron count rate over water. This concept has been tested by McJannet et al. (2014), Baatz et al. (2014) and Iwema et al. (2015) at different sites, but a general improvement in performance for quantifying soil moisture was not confirmed.

Recently, in this journal, Heistermann et al. (2021) demonstrated the feasibility of what they referred to as the estimation of a "single $N_0$". For this purpose, they used 18 CRNS sensors that were distributed as a cluster in an area of 1 km$^2$ in a prealpine catchment in Germany (Fersch et al., 2020). The study area was characterized by substantial landscape heterogeneity, including





grassland and mature forests, mineral and peat soils as well as locations close and distant to the groundwater table. Within that $1\,\mathrm{km}^2$, it was possible to use a single value of $N_0$ after the effects of different sensor sensitivities as well as hydrogen pools (biomass, soil organic matter, lattice water) were carefully accounted for. As for hydrogen pools, obviously, the difference between forest (above-ground dry biomass around $24\,\mathrm{kg/m}^2$) and grassland (around $0.2\,\mathrm{kg/m}^2$) was a dominant factor.

In this paper, we would like to revisit the idea of a general functional relationship as suggested by Zreda et al. (2012) and

Franz et al. (2013). This requires to account for the relative sensitivity of the neutron detector, for the effects of other hydrogen pools in the sensor footprint, as well as for the effects of geographic latitude, longitude, and altitude. To that end, we build upon Eq. 1 and combine it with well-established functional relationships and models, as outlined in section 3. We then use four recently published CRNS datasets in order to estimate a single value of $N_0$ to be applied across all sensors.

Yet, we would like to go one step further. While Franz et al. (2013) saw the value of a "universal calibration function" rather

for situations in which reference measurements of $\theta$ were unfeasible, we would like to ask whether omitting a local calibration could be an opportunity to avoid a fundamental source of uncertainty: the reference measurement of $\theta_\mathrm{cal}$. In order to answer that question, we analyse the propagation of errors for two contrasting calibration scenarios, local and general. Based on this uncertainty analysis, we will outline typical constellations under which one or the other option would be preferable, and give a rough assessment of the dominant sources of uncertainty.

## 2 Data

Recently, four major European CRNS datasets were published via Copernicus' journal Earth System Science Data.

In the **COSMOS-Europe** dataset, Bogena et al. (2022a) compiled neutron counts data and reference observations of $\theta_\mathrm{cal}$ as well as other variables (such as bulk density, soil organic carbon, lattice water, barometric pressure, air humidity and temperature) for 66 CRNS stations from 24 research institutions across Europe.

Moreover, three dedicated field campaigns were carried out by the Cosmic Sense research unit, a consortium of eight research institutions funded by the German Research Foundation (DFG). These campaigns all had in common the concept of dense CRNS clusters, meaning that a relatively large number of CRNS detectors (8-18) was operated within a relatively small area $(0.1 - 1\,\mathrm{km}^2)$. All three campaigns included extensive soil sampling to obtain reference measurements of $\theta_\mathrm{cal}$, but also soil bulk density, soil organic matter content, lattice water, and above-ground biomass for forested and non-forested areas. We will refer

to each of these datasets by the location of the campaign:

- **Fendt**: Fersch et al. (2020) published the results of a large campaign from May to July 2019 during which 18 CRNS detectors were continuously operated as a cluster within an area of $1\,\mathrm{km}^2$, the pre-Alpine upper Rott catchment in southern Germany. It is this dataset for which Heistermann et al. (2021) already demonstrated the feasibility of estimating one single $N_0$ for a large set of CRNS footprints.





– **Wüstebach**: a similar campaign with 15 CRNS was carried out from September to November 2020 in the $0.4\,\text{km}^2$
Wüstebach catchment (Eifel mountains in western Germany), which is governed by mature spruce forest, together with
a significant clear-cut area, at altitudes between 595 to 628 m (Heistermann et al., 2022)

   – **Marquardt**: recently, Heistermann et al. (2023) published a dataset of eight CRNS per $0.1\,\text{km}^2$ that were operated over
a period of three years in an agricultural research site in the lowlands of north-east Germany.

The four datasets are publicly available and comprehensively documented in the above references. Together, they include a
total of 107 CRNS stations. An important feature of this combined dataset is that it covers different spatial scales: while the
COSMOS-Europe dataset extends over Europe, the three dense CRNS clusters are distributed across Germany while each of
them, in turn, covers substantial heterogeneity at extents between 0.1 and $1\,\text{km}^2$. At these different scales (continental to local),
different factors are expected to govern the variability of neutron intensity: while the effect of the geomagnetic field might be
important at the continental scale, the effect of altitude might play a role at the regional scale whereas the heterogeneity of the
landscape with regard to different hydrogen pools might be dominant at the field or small catchment scale.

## 3 Methods

### 3.1 A general function for $\theta(N)$

The proposed general function for $\theta^G(N)$ builds on well-established community standards. In essence, we introduce various
terms to Eq. 1 which take into account the previously mentioned effects on epithermal neutron intensity. These terms either
multiplicatively scale the observed neutron intensity, or they additively represent other hydrogen pools as equivalents of soil
water. The resulting equation corresponds to Eq. 1 in Power et al. (2021), supplemented by the correction factor $f_s$:

$$\theta^G(N) = \left( \frac{a_0}{f_p \cdot f_h \cdot f_{\text{in}} \cdot f_b \cdot f_s \cdot \frac{N}{N_0} - a_1} - a_2 - \theta_g^{\text{OM}} - \theta_g^{\text{LW}} \right) \cdot \frac{\rho_b}{\rho_w} \tag{2}$$

The dimensionless multiplicative scaling factors $f$ represent the effects of barometric pressure ($f_p$), air humidity ($f_h$),
incoming neutron intensity ($f_{\text{in}}$), vegetation biomass ($f_b$), and detector sensitivity ($f_s$). $\theta_g^{\text{OM}}$ and $\theta_g^{\text{LW}}$ are the equivalents of
gravimetric soil water content resulting from soil organic matter and lattice water, respectively (in g/g).

   If we assume that Eq. 2 represents all relevant processes that affect the relationship between $\theta$ and $N$, the parameter $N_0$
should be the same in any location which meets this assumption (note that, in this study, we do not account for the presence
of snow, or for topographic shielding of cosmogenic neutrons in locations with complex and steep topography, see e.g. Dunne
et al., 1999; Balco, 2014; Schattan et al., 2019). The various components of Eq. 2 are detailed in the following:

$$f_p = f_p(p) = \exp\left( \frac{p - p_0}{L} \right) \tag{3}$$





$f_p$ was suggested by Zreda et al. (2012) and accounts for the effects of barometric pressure variations over time ($p$, in g/cm$^2$: barometric pressure at the time of the neutron measurement; $p_0$, in g/cm$^2$: arbitrary reference pressure, e.g the long term average of $p$ at the measurement site, or the standard pressure at the altitude of the station; $L$, in g/cm$^2$: mass attenuation

length for high-energy neutrons, set to a constant value of 131.6 g/cm$^2$). In theory, $L$ depends on the geomagnetic location, but Bogena et al. (2022b) found no significant variation across Europe. In this study, we assume that any remaining effects of cut-off rigidity will be accounted for by the correction function $f_{\text{in}}$ (see Eq. 5).

$$f_h = f_h(h) = 1 + \alpha \cdot (h - h_0) \tag{4}$$

Rosolem et al. (2013) suggested $f_h$ in order to account for the temporal variation of the absolute humidity of the air ($h$,

in g/m$^3$) from an arbitrary reference $h_0$ (here the temporal average of $h$ at the measurement site in Marquardt, yielding $h_0 = 7.5$ g/m$^3$), and determined the value of $\alpha$ as 0.0054 m$^3$/g by means of neutron simulations.

$$f_{\text{in}} = f_{\text{in}}(\phi, \lambda, z, I) = f_{\text{in}}^t(I) \cdot f_{\text{in}}^s(\phi, \lambda, z) \quad \text{with} \quad f_{\text{in}}^t(I) = \frac{I_0}{I}, \quad f_{\text{in}}^s(\phi, \lambda, z) = \frac{\xi(\phi_0, \lambda_0, z_0)}{\xi(\phi, \lambda, z)} \tag{5}$$

$f_{\text{in}}$ accounts for the temporal ($f_{\text{in}}^t$) and spatial ($f_{\text{in}}^s$) variation of incoming high-energy neutrons. Typically, CRNS-related studies only consider $f_{\text{in}}^t$ by relating the secondary neutron intensity $I$ (in cph) observed by one of the monitors in the neutron

monitor database (NMDB) to an arbitrary reference intensity $I_0$ (here the average of $I$ at the monitor between 2009 and 2023). For most CRNS applications in Europe, the neutron monitor at Jungfraujoch ("JUNG" in the NMDB) is chosen for that purpose, and we do the same in this study.

The spatial variation $f_{\text{in}}^s$ of incoming neutrons is typically not considered, but for a general relationship $\theta(N)$, it becomes crucial. $f_{\text{in}}^s$ is a function of the geomagnetic field of the Earth (which varies with longitude $\lambda$ and latitude $\phi$, both in decimal

degrees), and the attenuation by the atmosphere which, in turn, is a function of altitude ($z$, in m a.s.l.). For this study, we use the PARMA model (Sato, 2015) to simulate $f_{\text{in}}^s$ in a consistent way. While the PARMA model covers a wide range of particles and energy levels, the value $\xi(\lambda, \phi, z)$ in Eq. 5 corresponds to the average of simulated neutron intensities at energies from 1 to $10^5$ eV. That way, we directly represent the effect of location ($\lambda, \phi, z$) on the average epithermal neutron intensity. Although the PARMA model is able to account for some aspects of the temporal variation of incoming neutrons (e.g. solar cycles), we use

an arbitrary reference date (May 31, 2019) as input since the temporal variation of incoming neutron intensity across various times scales is represented by $f_{\text{in}}^t$. In order to obtain a scaling factor $f_{\text{in}}^s$, we scale $\xi$ at any location ($\lambda, \phi, z$) by $\xi$ at an arbitrary location defined by $\lambda_0$=12.97°, $\phi_0$=52.47°, and $z_0$=40 m). This arbitrary location corresponds to the research site in Marquardt (about 10 km southwest of Berlin, Germany, see Heistermann et al., 2023).

$$f_b(t) = \frac{1}{1 - 0.009 \cdot \text{AGB}} \tag{6}$$





**Table 1.** Average detector efficiencies (sensitivities) observed for the most common detector models, all from Hydroinnova Ltd, USA (94 out of the 107 detectors in the combined dataset are of one of these types). Note that the column "sensitivity" shows the inverse of $f_s$: the lower the count efficiency of the detector, the higher the value of $f_s$.

| Detector type | Sensitivity ($f_s^{-1}$) |
|---|---|
| Calibrator | 1.0 |
| CRS-1000 | 0.452 |
| CRS-1000B | 0.668 |
| CRS-2000 | 0.871 |
| CRS-2000B | 1.147 |

$f_b$ accounts for the effect of vegetation biomass on neutron count rates: following the empirical analysis of Baatz et al. (2015), we assume that epithermal neutron count rates are reduced by 0.9 % for every kg of dry above-ground biomass per m$^2$ (AGB).

$$f_s = \frac{N_{\text{ref}}}{N} \tag{7}$$

The sensitivity factor $f_s$ accounts for the detector efficiency and is used to scale observed neutron intensities $N$ to the
intensity $N_{\text{ref}}$ that would have been observed by an arbitrary reference detector. As such reference detector, we chose a so-called "calibrator" probe (manufactured by Hydroinnova, two counter tubes based on $^3$He gas). During the above-mentioned campaigns in Fendt, Wüstebach and Marquardt, we collocated such a calibrator with various types of CRNS sensors over longer periods (typically several days) in order to obtain $f_s$. For sensors which are missing a calibrator collocation (most sensors in the COSMOS-Europe dataset, a few sensors in the other three datasets), we assumed the average value of $f_s$ for
the corresponding detector type (see Tab. 1). One should keep in mind, though, that sensitivity might slightly vary between instruments of the same type (Schrön et al., 2018, found variations of 1-3 % in an intercomparison study that included nine CRS-1000 instruments). Another way to replace a calibrator measurement is to collocate a sensor with another sensor for which $f_s$ is known ("cross-calibration"). In case neither a direct reference measurement nor an average $f_s$ for a detector type nor cross-calibration is an option, $f_s$ remains unknown and Eq. 2 cannot be applied. While this applies to any variable in Eq. 2
required for the general calibration strategy, the quantification of $f_s$ might constitute a particular challenge in case a suitable reference is unavailable to the user, e.g. when a new type or brand of CRNS detector is introduced. Within the set of 107 CRNS sensors from our four datasets, we were able to retrieve $f_s$ for 100 locations.

$$\theta_g^{\text{OM}} = 0.556 \cdot \text{OM} \tag{8}$$





Following McJannet et al. (2014), $\theta_g^{\mathrm{OM}}$ is 0.556 times the organic matter content (OM, g/g), based on the stoichiometry of
cellulose. Finally, $\theta_g^{\mathrm{LW}}$ can be taken directly from measurements.

## 3.2 Data filtering and processing

For the estimation of $N_0$ in Eq. 2, we discarded a number of CRNS locations and calibration dates from the COSMOS Europe
dataset, namely locations or calibration dates for which

- the sensitivity factor $f_s$ was unknown and could not be inferred from the sensor type;

- the soil sampling data for calibration was insufficient (e.g. less than 18 profiles, only surface sampling, missing bulk
  density, etc.);

- the sensor was placed in or close to a forest, but no biomass estimates were available to us (which effectively applies to
  most forest locations in the COSMOS Europe dataset).

From the COSMOS-Europe location with ID JEC001 (Jena), we randomly selected four out of a total of 30 calibration dates
in order to avoid that the location was over-represented in the calibration dataset.

After filtering, 75 CRNS locations with a total of 104 calibration dates were still available for analysis. Based on the pub-
lished datasets, we derived the required parameters of Eq. 2 for each location and calibration date. For bulk density, soil organic
matter content, lattice water, above-ground dry biomass, and volumetric soil moisture, we obtained weighted average values
by applying the weighting function provided by Schrön et al. (2017).

## 3.3 Error propagation

As pointed out in the introduction, we aim to compare two contrasting application scenarios with regard to the resulting
uncertainty for $\theta(N)$:

1. the use of a **general calibration** of $\theta^G(N)$, Eq. 2, for which we need to determine a wide range of location-specific
   parameters, but can estimate and then apply a location-independent estimate of $N_0$.

2. the **local calibration** of $N_0$ in Eq. 1 for which we require a local reference measurement of $\theta_{\mathrm{cal}}$ and an estimate of the
   soil bulk density $\rho_b$ within the sensor footprint.

Assuming independent variables and normally distributed errors, we can apply Gaussian error propagation for both scenarios.
This approach has been followed by other studies on neutron counts Weimar et al. (2020); Schrön et al. (2021), on neutrons
and bulk density (Jakobi et al., 2020), and on neutrons, $N_0$, meteorological parameters, and sampling tube geometries (Gugerli
et al., 2019).

For the **general calibration**, the uncertainty of $\theta^G(N)$ is obtained by propagating the errors of the following variables: the
sensitivity factor $f_s$, the observed neutron intensity $N$, the estimated value of $N_0$, the above-ground dry biomass AGB, the





organic matter content OM, the lattice water content $\theta_{\mathrm{LW}}$, and the soil bulk density $\rho_b$. For the sake of simplicity, the effects of the correction factors $f_p$, $f_h$, and $f_{\mathrm{in}}$ are not included: the underlying measurement uncertainties of pressure, humidity,

incoming neutron intensity at Jungfraujoch are considered as relatively small while the uncertainty of $f_{\mathrm{in}}^s$ as derived from the PARMA model is difficult to quantify. Moreover, $f_p$, $f_h$, and $f_{\mathrm{in}}$ would be applied in the local calibration scenario, too, so both scenarios were similarly affected.

In summary, the following equation describes the uncertainty of $\theta^G(N)$ in terms of its standard deviation $\sigma_{\theta G}$ (in m³/m³):

$$\sigma_{\theta G} = \sqrt{\sum_x \left(\frac{\partial \theta_G}{\partial x}\right)^2 \sigma_x^2} \quad \text{for} \quad x \in \{f_s, \mathrm{AGB}, N, N_0, \mathrm{OM}, \theta_{\mathrm{LW}}, \rho_b\} \tag{9}$$

where any $\sigma$ denotes the error of the respective variable $x$.

For the error propagation in the **local calibration** scenario, we used Eq. 1 to derive an expression, $\theta^L(N)$, which provides the CRNS-based SWC as a function of the local calibration measurements ($N_{\mathrm{cal}}$, $\theta_{\mathrm{cal}}$) and of the observed neutron intensity $N$ at any point in time. For this purpose, we first insert $N_{\mathrm{cal}}$ and $\theta_{\mathrm{cal}}$ into Eq. 1, solve for $N_0$ (i.e. calibrating the local $N_0$), and then insert the resulting term again to Eq. 1 (i.e. applying the locally calibrated $N_0$).

Typically, the neutron intensities $N$ and $N_{\mathrm{cal}}$ are corrected for the temporal variation of pressure, humidity and incoming neutrons. We summarize these correction factors as $\tau = f_p \cdot f_h \cdot f_{\mathrm{in}}$ which corresponds to the period during which $N$ was observed, while $\tau_{\mathrm{cal}}$ is the corresponding correction factor for $N_{\mathrm{cal}}$. Altogether, we obtain:

$$\theta^L(N) = \left(a_0 \cdot \left(\frac{\tau \cdot N}{\tau_{\mathrm{cal}} \cdot N_{\mathrm{cal}}} \cdot \left(\frac{a_0}{\theta_{\mathrm{cal}} \cdot \frac{\rho_w}{\rho_b} + a_2} + a_1\right) - a_1\right)^{-1} - a_2\right) \cdot \frac{\rho_b}{\rho_w} \tag{10}$$

We then propagated the errors of $N$, $N_{\mathrm{cal}}$, $\theta_{\mathrm{cal}}$ and $\rho_b$ to $\theta^L(N)$ to obtain the corresponding error $\sigma_{\theta L}$ (in m³/m³), while
neglecting the errors of $f_p$, $f_h$, and $f_{\mathrm{in}}$ as explained above:

$$\sigma_{\theta L} = \sqrt{\frac{\partial \theta^L}{\partial N}^2 \sigma_N^2 + \frac{\partial \theta^L}{\partial N_{\mathrm{cal}}}^2 \sigma_{N_{\mathrm{cal}}}^2 + \frac{\partial \theta^L}{\partial \theta_{\mathrm{cal}}}^2 \sigma_{\theta_{\mathrm{cal}}}^2 + \frac{\partial \theta^L}{\partial \rho_b}^2 \sigma_{\rho_b}^2} \tag{11}$$

For the local calibration scenario, it could also be an option to use Eq. 2 instead of the simplified Eq. 1. This would require to quantify all parameters as precisely as possible (particularly the additive offsets $\theta_{\mathrm{OM}}^g$ and $\theta_{\mathrm{LW}}^g$) and eventually to estimate the local $N_0$. Ideally, this approach would make estimates of $N_0$ more consistent between different locations, corresponding to an
intermediate between the local and the general calibration strategy. In our study, however, we decided to limit the analysis to a "purely local" approach, where the simpler Eq. 1 is used in order to avoid the introduction of additional sources of uncertainty. The strength of this approach is that the uncertainties of all other parameters could be effectively lumped into the estimation of $N_0$. Future uncertainty analyses, however, might decide to include at least the offset terms $\theta_{\mathrm{OM}}^g$ and $\theta_{\mathrm{LW}}^g$ in the evaluation of the local calibration approach.

The required partial derivatives of $\theta^G$ and $\theta^L$ are provided in the supplementary to this technical note.





## 4 Results and discussion

### 4.1 A general function for $\theta(N)$

As pointed out in Sect. 3.2, a total of 75 CRNS locations and 104 calibration dates remained after applying a set of filtering rules. Based on this subset, $N_0$ in Eq. 2 was determined by minimizing the mean absolute error (MAE) between $\theta^G(N_{\mathrm{cal}})$

and $\theta_{\mathrm{cal}}$, yielding $N_0 = 2306$ cph and MAE=0.075 m³/m³. Note that this value of $N_0$ has no fundamental physical meaning; it is a result of our general calibration framework in which all neutron count rates are scaled to arbitrary references for sensitivity (Hydroinnova's calibrator), geographic location (Marquardt), and conditions without vegetation. Fig. 1 illustrates the calibration results.

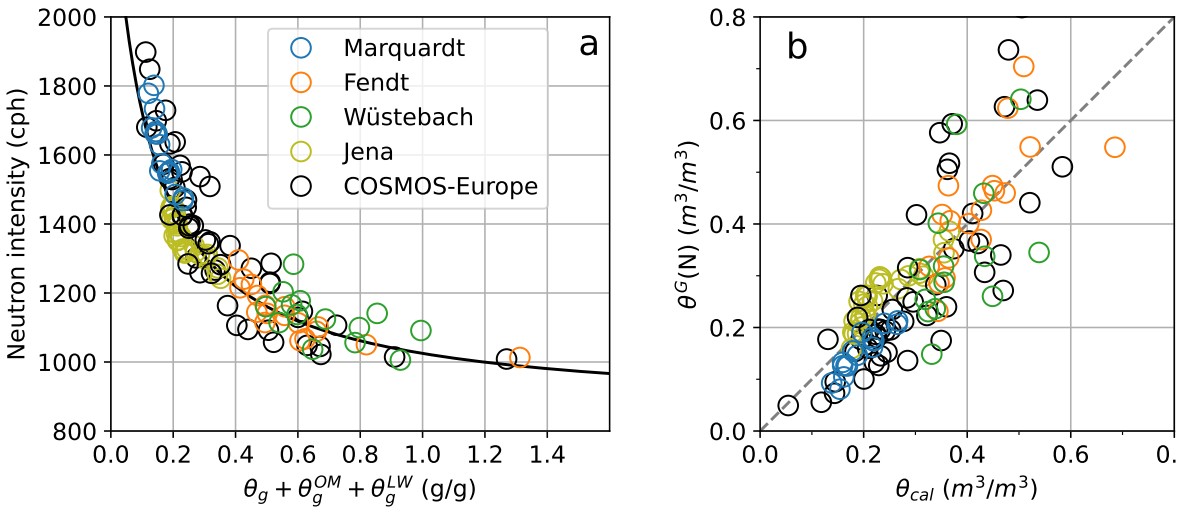

**Figure 1. a)** Scaled neutron intensity (all multiplicative factors $f$ in Eq. 2 applied) plotted over "apparent gravimetric SWC", i.e. the sum of gravimetric SWC and the equivalents of SWC resulting from soil organic matter and lattice water. The line shows Eq. 2, solved for the scaled neutron intensity; **b)** $\theta^G(N)$ (i.e. CRNS-based estimate of SWC) over $\theta_{\mathrm{cal}}$ (i.e. SWC obtained from soil sampling) for all analysed CRNS locations and calibration dates. For Jena (location JEC001 in COSMOS-Europe), we show the estimates for all available calibration dates, not only the four ones selected for calibration.

Fig. 1a shows that Eq. 2 captures the relationship between neutron intensity and apparent soil moisture fairly well, although

some points show substantial deviations from the function line. This specifically applies to some points from the COSMOS-Europe (black circles) and the Wüstebach dataset (green circles). For the latter, the uncertainty of above-ground biomass and the related scaling factor $f_b$ is assumed to be high. Furthermore, there appears to be an underestimation of soil moisture by $\theta^G(N)$ for dry conditions below 0.2 g/g. This is in line with recent findings by Köhli et al. (2020) who demonstrated that the original equation proposed by Desilets et al. (2010) is not steep enough for dry conditions, and proposed a functional



relationship to address this issue. While it should be straightforward for future studies to apply the presented findings to any new functional relationship between $N$ and $\theta$, such as the one proposed by Köhli et al. (2020), we will stick, in the present analysis, to the Desilets equation as it is still the community standard.

Looking at how $\theta^G$ corresponds to $\theta_{\mathrm{cal}}$ from soil sampling (Fig. 1b), we note a much higher level of scatter. This is plausible, and also consistent with previous findings, because the retrieval of volumetric soil moisture estimates - in contrast to the 235 "apparent gravimetric soil moisture" - introduces additional uncertainty, most notably from the estimation of soil bulk density. Accordingly, Franz et al. (2013) had already noted that "[...] accurate spatial estimates of volumetric water content may be difficult to obtain because of a large uncertainty in the determination of soil bulk density [...]" (corresponding uncertainty analyses were also carried out by e.g. Jakobi et al., 2020; Iwema et al., 2021).

Despite the scatter in Fig. 1b, the estimation of $N_0$ from this dataset is robust. Via bootstrapping, we determined the standard 240 deviation of $N_0$ to be 15 cph, which is less than 1 %. This is a result of the large number of calibration locations and dates. Obviously, $N_0$ would vary substantially if we estimated it individually for each calibration date (i.e. for each point in Fig. 1a).

Surely, we would like to know the reasons behind the scatter in Fig. 1b. The honest answer, however, is that we cannot tell. Each circle in the plot could tell a different story of coinciding uncertainties.

The error in the $x$-dimension relates to what we informally refer to as "ground truth", although the actual level of *truth* in 245 $\theta_{\mathrm{cal}}$ remains difficult to determine. All we know is that numerous errors might accumulate along the way, e.g. the measurement error of $\theta$ at a single point (possibly systematic, depending on technology), the effects of limited sample size in combination with the limited horizontal and vertical representativeness of each measurement, or the uncertainty of the horizontal and vertical weighting functions.

In the y-dimension, all parameters in Eq. 2 come with considerable uncertainty. However, we expect the parameters derived 250 from soil sampling in the CRNS footprint (soil bulk density, organic matter content) as well as the above-ground biomass (specifically in forests) as particularly uncertain and not straightforward to quantify. In contrast, the stochastic uncertainty of $N$ itself is well-known (see Sect. 4.2).

And so a fundamental question arises from our ignorance of the *specific* reasons behind each mismatch in Fig. 1b: Should we trust our general calibration function, or should we calibrate locally? If we considered $\theta_{\mathrm{cal}}$ to be the dominant source of 255 uncertainty, we would go with the general calibration. If we considered the parameters in Eq. 2 to govern the uncertainty of $\theta(N)$, we would prefer a local calibration.

In order to better understand the trade-offs between both options, the next section will take a closer look at the corresponding propagation of errors.

## 4.2 Error propagation

260 The error propagation to estimate the error of $\theta(N)$ for the two calibration strategies, local or general, was outlined in Sect. 3.3. Fig. 2 shows selected results for the local calibration (Eq. 10). The combinations and ranges of parameters that were used to create the figure are, to some extent, exemplary choices. Later in this paper, we will refer to an online tool in which potential users can explore specific parameter combinations on their own.



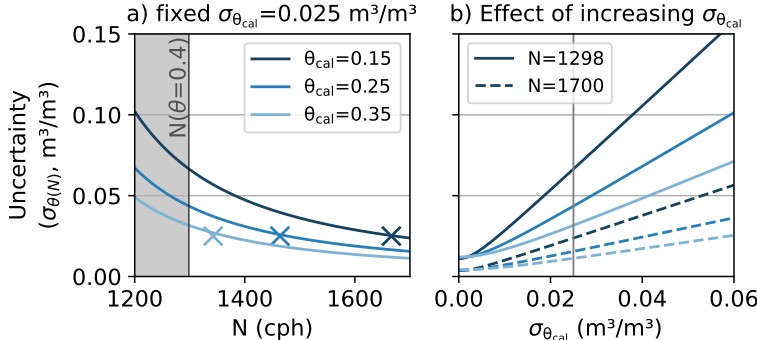

**Figure 2.** Standard deviation of $\theta^L(N)$ in case a local value of $N_0$ is estimated from a local reference measurement of SWC, see scenario 1 in Tab. 2; **a)** dependence of the error on the wetness conditions during calibration (i.e. $\theta_{\text{cal}}$), assuming an arbitrarily fixed error of $\sigma_{\theta_{\text{cal}}}$=0.025 m$^3$/m$^3$. The "x" markers highlight the neutron intensity during calibration ($N_{\text{cal}}$) for the SWC corresponding to $\theta_{\text{cal}}$; **b)** error for increasing values of $\sigma_{\theta_{\text{cal}}}$ at two neutron intensity levels (dashed lines for dry conditions, solid lines for wet conditions); colors are explained by the legend in subplot a. The vertical grey line corresponds to the value of $\sigma_{\theta_{\text{cal}}}$, which was fixed for subplot a.

While we have to make assumptions about the standard deviation (or error) of most involved parameters, the uncertainty $\sigma_N$ of the neutron count rate $N$ (in cph) is a result of the stochastic nature of the counting process and amounts to $\sqrt{N}/\sqrt{\Delta t}$ for an integration period of $\Delta t$ (in hours; in this study, we always use $\Delta t$=24 h). As a result, the *relative* uncertainty of $N$ increases with decreasing $N$ (and hence increasing $\theta$). This fact is well known (see e.g. Francke et al., 2022), and clearly visible in Fig. 2a. The same figure, however, shows that the increase of $\sigma_{\theta(N)}$ with decreasing $N$ very much depends on the wetness conditions under which the calibration was carried out: for the same value of $N$, the error of $\theta(N)$ is higher in case the calibration was carried out under drier conditions. While this behaviour is plausible, it might appear surprising at first, and was, to our knowledge, not described before. The same error of $\theta_{\text{cal}}$ will be amplified under wet conditions when the calibration were carried out under dry conditions while the error will be attenuated under dry conditions if the calibration was carried out under wet conditions. This is only partly due to the fact that the same value of $\sigma_{\theta_{\text{cal}}}$ implies different relative errors under wet and dry conditions. The more important effect is the different slope of the Desilets function under wet and dry conditions.

Fig. 2b shows the effect of increasing values of $\sigma_{\theta_{\text{cal}}}$, again for the three different calibration conditions. For each of the three $\theta_{\text{cal}}$, we show the error for two values of $N$ - one for wet (solid line) and one for dry conditions (dashed line). The solid dark blue line gives us a kind of worst case scenario, with $\sigma_\theta$ exceeding values of 0.15 m$^3$/m$^3$.However, we should keep in mind that SWC will typically vary between wilting point and field capacity (or porosity at the maximum) which spans a limited dynamic range of SWC for most soils.

Before turning to the general calibration function, we should note that the error of soil bulk density ($\sigma_{\rho_b}$) does not propagate to $\theta^L(N)$ (not shown in the figure). The reason is that the influence of bulk density basically cancels out as it appears in the





enumerator and denominator of Eq. 10. This is different for the general calibration function, for which the results are shown in Fig. 3.

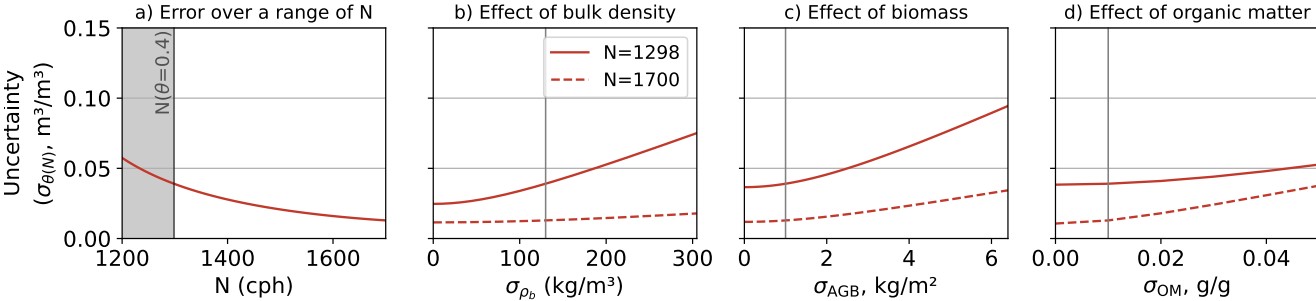

**Figure 3.** Standard deviation of $\theta(N)$ in case of a general calibration. **a)** error of $\theta^G(N)$ for parameters as shown in Tab. 2, scenario 1.**b-d**: same parameters, except for the errors of bulk density **(b)**, above-ground biomass **(c)** and organic matter content **(d)**. The vertical grey lines mark the values of the corresponding error that were used to create subplot (a).

The layout of that figure is similar to Fig. 2. As the general calibration strategy relies on a single location-independent value

of $N_0$, no different calibration conditions need to be compared here. However, more parameters can potentially propagate their errors. Fig. 3a shows the error for a parameter combination that is referred to as scenario 1 in Tab. 2. The resulting curve is similar to the light blue curve in Fig. 2a (high wetness during calibration). Figs. 3b-d illustrate how the error of $\theta^G(N)$ changes with the errors of bulk density, above-ground biomass and soil organic matter content. Specifically under wet conditions ($N = 1298$ cph, corresponding to $\theta = 0.4\,\mathrm{m}^3/\mathrm{m}^3$), large errors in the estimation of soil bulk density or biomass in

the sensor footprint will substantially increase the error of $\theta^G(N)$, which is to be expected. For biomass, though, it must be emphasized that the upper range of errors in the estimation of biomass are only expected to occur in forested areas (where, at least for temperate conditions, total above-ground dry biomass typically ranges between 10 and $40\,\mathrm{kg}/\mathrm{m}^2$). In grassland or cropland, however, above-ground dry biomass will mostly not exceed $1\,\mathrm{kg}/\mathrm{m}^2$, so that the corresponding estimation error is expected to remain much lower.

We would like to come back to our question which strategy, local or general, is recommended in terms of minimizing the error of $\theta(N)$. The above results already suggest that there is no general answer to that question. Instead, the answer depends on the specific combination of parameters and errors that we expect to govern the sensor's response and the data sampled within its footprint. Unfortunately, we usually do not know these values, particularly in case of soil or biomass sampling where we are dealing with limited sample sizes, or, even worse, a lack of representativeness. But while we might not know the exact

errors we are dealing with, we might at least be able to make an educated guess, to narrow down the ranges, or give maximum error estimates. For instance, as mentioned above, we can be very certain that the error in our above-ground biomass estimate will not exceed $1\,\mathrm{kg}/\mathrm{m}^2$ in a grassland or cropland location. Based on the relationships shown in Fig. 3, CRNS users might also decide to increase their sampling efforts (for any variable such as $\theta_{\mathrm{cal}}$, $\rho_b$, or AGB) until they can be confident that the error of that variable remains within a desired range.





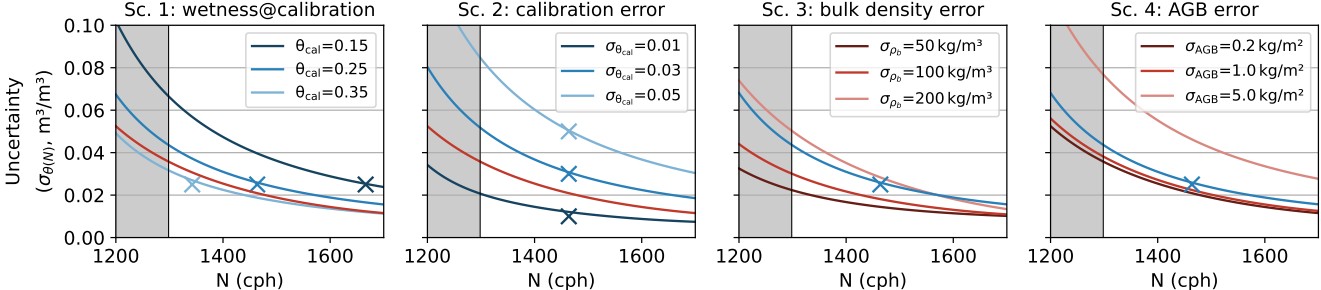

**Figure 4.** Similar to Figs. 2a and 3a, but for four different scenarios ("Sc.", see Tab. 2 for the definition of the scenarios). Blueish colors correspond to the local, reddish colors to the general calibration strategy.

**Table 2.** Parameter combinations for the scenarios evaluated in this study. Note that neither the general calibration (Eq. 2) nor the local one (10) use all parameters contained in one row of the table (e.g. $\theta_{cal}$ is only used for local calibration, not for general).

| Scenario | $\theta_{cal}$ (m³/m³) | $\sigma_{\theta_{cal}}$ (m³/m³) | $\rho_b$ (kg/m³) | $\sigma_{\rho_b}$ (kg/m³) | AGB (kg/m²) | $\sigma_{AGB}$ (kg/m²) | OM (g/g) | $\sigma_{OM}$ (g/g) | $f_p, f_h, f_{in}, f_s$ – | $\Delta t$ (h) |
|---|---|---|---|---|---|---|---|---|---|---|
| default | 0.25 | 0.025 | 1300 | 130 | 2 | 0.2 | 0.06 | 0.01 | 1 | 24 |
| 1 | 0.15, 0.25, 0.35 | | | | | | | | | |
| 2 | | 0.01, 0.03, 0.05 | | | | | | | | |
| 3 | | | | 50, 100, 200 | | | | | | |
| 4 | | | | | | 0.2, 1, 5 | | | | |

Fig. 4 directly compares the two calibration strategies for a few selected scenarios (Tab. 2), in order to convey some basic guidance. As already shown above, for the local calibration, $\theta(N)$ degrades substantially with increasing errors of $\theta_{cal}$ (scenario 2). For the general calibration, the uncertainty of $\theta(N)$ is governed by the errors of bulk density and biomass (see scenarios 3 and 4). While this qualitative behaviour is unsurprising, the strength of such a visualization is that it quantitatively contrasts the results for potential applications contexts. That way, it becomes obvious, for example, that the general calibration outperforms

the local one for agricultural landscapes (low biomass error) and moderate errors in bulk density and $\theta_{cal}$, while the local calibration is clearly preferable in forest environments, unless very reliable biomass estimates are available.

For users who would like to explore how the two calibration strategies compare in their specific application context, we provided an interactive online tool: https://cosmic-sense.github.io/local-or-global.





## 5    Conclusions

We tested a general functional relationship $\theta^G(N)$ to estimate SWC from observed neutron intensities, without the need for a local calibration. $\theta^G(N)$ is based on the widely used Desilets function, and takes into account various variables which govern the neutron intensity observed in a specific location. To calibrate and test $\theta^G(N)$, we used four recently published datasets with a total of 75 CRNS locations and 104 calibration dates. This constitutes the most comprehensive analysis on CRNS calibration conducted to far.

Apart from accounting for the local effects of vegetation, soil organic matter, lattice water, and bulk density, two features were essential to achieve the desired level of generalization, i.e. to estimate one single value of $N_0$ across all CRNS locations:

- Accounting for detector efficiency was possible thanks to comprehensive instrumental efforts undertaken during campaigns in the context of the three dense CRNS clusters. In these campaigns, CRNS sensors were systematically collocated with a so-called "calibrator" probe so that we were able to determine either specific sensitivity factors for individual neu-
tron detectors or average sensitivity factors for the most common types of detectors, relative to that "calibrator" probe. Evidently, we cannot apply $\theta^G(N)$ if the relative sensitivity of the neutron detector is unknown. This is a fundamental caveat, although prospective research might find ways to address this issue, e.g. by considering neutron detectors as nodes in a topological network so they could be cross-calibrated across multiple edges of such a network, or by simulating response functions of CRNS detector designs (see e.g. Köhli et al., 2018).

- We used the PARMA model (Sato, 2015) in order to account for the spatial variability of incoming neutron intensity (relative to a reference location), as a function of geographic latitude and longitude as well as terrain altitude. While the PARMA model is well-established, other such models exist, and future research might explore the potential sensitivity of the neutron intensity scaling on the choice of the model.

Altogether, we assume having considered the most relevant processes and variables (except for the presence of snow and for
topographic shielding). On average, the general calibration function fits fairly well the weighted average of locally measured SWC per CRNS footprint, specifically for the apparent gravimetric SWC. Looking at volumetric soil moisture, though, the mean absolute error between $\theta^G(N)$ and the calibration reference $\theta_{\mathrm{cal}}$ amounts to $0.075\,\mathrm{m^3/m^3}$ – which is quite substantial.

If we trust the general structure of our function $\theta^G(N)$, this error can have two basic sources: the uncertainty of $\theta_{\mathrm{cal}}$ and the uncertainty of the parameters in $\theta^G(N)$. We expect both parts to be error-prone, and cannot quantify either one with confidence.
Interestingly, though, we only become aware of these errors in case we apply a general calibration. If we calibrate $N_0$ locally, we will, by definition, force $\theta^L(N_{\mathrm{cal}})$ and $\theta_{\mathrm{cal}}$ to be equal at the date of calibration.

Most CRNS users consider opting out of local $N_0$ calibration only if local reference measurements of SWC are impossible, e.g. due to stony soils, restricted access, or in the case of CRNS roving. Based on the results of this study, we recommend to consider both calibration options, local and general, and to try weighing the uncertainty of the one against the other. We used
Gaussian error propagation for that purpose. Unsurprisingly, the error of $\theta^L(N)$ is governed by the error of the calibration reference $\theta_{\mathrm{cal}}$. However, it was interesting for us to note how the propagation of this error depends on the value of $\theta_{\mathrm{cal}}$ itself:



if the calibration were carried out under dry conditions, the error would grow substantially under wet application conditions. Please note, however, that we only analysed the error of the local calibration scenario in the case of a single calibration date (while it is recommended to carry out multiple calibration campaigns, this is not yet common practice). The overall error

of $\theta^L(N)$ is expected to decrease for multiple calibration dates, and future studies should aim to explicitly represent the corresponding error propagation.

The error of $\theta^G(N)$, in turn, is governed by the errors of vegetation biomass and bulk density. Altogether, there is no general answer as to which calibration option is preferable. Based on our results, though, users should be aware that even in the presence of local calibration measurements, actually *applying* a local calibration might not necessarily be the best option.

Instead, we provide an interactive tool so that users can weigh the options in their specific application context, or decide how much additional sampling efforts are required to reduce the uncertainty of either calibration option.

We would like to emphasize that our formulation of a general calibration function should be considered as a mere suggestion. Other functions might be better suited, either for the overall relationship between $\theta$ and $N$ (such as the one provided by Köhli et al., 2020, specifically under dry conditions), or for the individual components that are required to scale the observed neutron

intensities (e.g. to account for vegetation biomass or the spatial variability of the incoming neutron flux). The resulting value of $N_0$ will depend much on the specific scaling techniques, so our value of 2306 cph should not be over-interpreted. Rather than to provide a universal calibration function with a universal $N_0$, the key lesson of this study is that it is worth using a multiplicity of locations for calibrating any $\theta(N)$ relationship, even if you are interested in only one specific location yourself. The CRNS datasets for such analyses are openly available to everyone, so other ideas can be explored.

*Code and data availability.* The four CRNS datasets used for this study were published via Copernicus' Earth System Science Data (Fersch et al., 2020; Heistermann et al., 2022; Bogena et al., 2022a; Heistermann et al., 2023). The PARMA model is openly available in the form of an Excel application (EXPACS), and as a C++ code, see https://phits.jaea.go.jp/expacs. We provide an interactive online tool at https://cosmic-sense.github.io/local-or-global. The corresponding JavaScript code, together with a juypter notebook for the analysis, is available via https://github.com/cosmic-sense/local-or-global.

*Author contributions.* MH and TF conducted the analysis and wrote the manuscript, MS and SO co-authored the manuscript.

*Competing interests.* The authors declare they have no competing interests.

*Acknowledgements.* This research was funded by the Deutsche Forschungsgemeinschaft (DFG, German Research Foundation) – research unit FOR 2694 "Cosmic Sense", project number 357874777.



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
