# Peer review of "Technical Note: revisiting the general calibration of cosmic-ray neutron sensors to estimate soil water content"

_Hydrology and Earth System Sciences, 2023_

## Author Comment (AC3)

**Author Response to Referee #2**

**Technical Note: revisiting the general calibration of cosmic-ray neutron sensors to estimate soil water content**

Maik Heistermann et al.

*Hydrol. Earth Syst. Sc.,* `doi:10.5194/hess-2023-169`
* * *
**RC:** *Referee Comment*,      AR: *Author Response*,      ☐ Manuscript text

Dear referee,

thank you very much for your positive response, and for the time and effort spent to examine the manuscript.

The comments are very useful and will be comprehensively considered in the revised version of the manuscript. Please find a point-by-point reply below.

Kind regards,
Maik Heistermann
(on behalf of the author team)

**Comments and responses**

**RC:** *[...] Line 60: Sentence is a bit clunky - suggest reword to "This requires the user to account for the relative sensitivity of the neutron detector, the effects of other hydrogen pools in the sensor footprint, and the effects of geographic latitude, longitude, and altitude"*

AR: The suggestion will be implemented.

**RC:** *Line 130: An alternative method for correcting for spatial and altitudinal variability has been published by McJannet and Desilets (2023) and it may be useful here as it will allow user to apply even more widely to areas outside of Europe.*

AR: This suggestion is in line with a comment made by referee #1, and we agree that McJannet and Desilets (2023) should be mentioned in the context of this study. The best opportunity to do this is, in our view, in the conclusions section after line 333 of the preprint. There, we already stated:

> While the PARMA model is well-established, other such models exist, and future research might explore the potential sensitivity of the neutron intensity scaling on the choice of the model.

In the context of this paragraph, we will more specifically refer to alternative approaches, including McJannet and Desilets (2023), but also e.g. Hawdon et al. (2014) and the scaling approach used in the original COSMOS

paper by Zreda et al. (2012). The latter was based on the method that had been presented by Desilets and Zreda (2003). Altogether, the above paragraph should become:

> While the PARMA model is well-established, its application in this study remains a subjective and exemplary choice. Other similar models exist (e.g. Desilets and Zreda, 2003; Hawdon et al., 2014; or McJannet and Desilets, 2023), and future research should aim to explore the potential sensitivity of the neutron intensity scaling to the choice of the model and the consistency of the resulting soil moisture estimates.

**RC:** *Line 145 suggest changing to "For our reference detector we chose a so called..."*

AR: The suggestion will be implemented.

**RC:** *Line 334: suggest "Altogether, we assume that we have considered the most relevant processes and variables..."*

AR: The suggestion will be implemented.

**RC:** *Line 335: suggest "On average, the general calibration function fits the weighted average of locally measured SWC per CRNS footprint fairly well, specifically for the apparent gravimetric SWC".*

AR: The suggestion will be implemented.

**RC:** *Line 340: suggest "Interestingly, though, we only become aware of these errors in the case where we apply a general calibration..."*

AR: The suggestion will be implemented.

**RC:** *Line 344: suggest "Based on the results of this study, we recommend considering both calibration options, local and general, and weighing the relative uncertainty of the one against the other"*

AR: The suggestion will be implemented.

**References**

Desilets, D. and Zreda, M. (2003): Spatial and temporal distribution of secondary cosmic-ray nucleon intensities and applications to in-situ cosmogenic dating, Earth Planet. Sc. Lett., 206, 21–42.

Hawdon, A., McJannet D., Wallace J. (2014) Calibration and correction procedures for cosmic-ray neutron soil moisture probes located across Australia. Water Resources Research, 50(6), 5029–5043.

McJannet, D. L., Desilets D. (2023): Incoming Neutron Flux Corrections for Cosmic-Ray Soil and Snow Sensors Using the Global Neutron Monitor Network. Water Resources Research, 59(4), e2022WR033889.

Zreda, M., Shuttleworth, W. J., Zeng, X., Zweck, C., Desilets, D., Franz, T., and Rosolem, R. (2012): COSMOS: the COsmic-ray Soil Moisture Observing System, Hydrol. Earth Syst. Sci., 16, 4079–4099.

---

## Author Response (AR1)

**Author Response Letter**

**Technical Note: revisiting the general calibration of cosmic-ray neutron sensors to estimate soil water content**

Maik Heistermann et al.

*Hydrol. Earth Syst. Sc.,* `doi:10.5194/hess-2023-169`
* * *
**EC:** *Editor Comment*,  **RC:** *Referee Comment*,  AR: *Author Response*,  ☐ Manuscript text

Dear editor, dear referees,

thank you very much for the positive and constructive feedback, and for your time and resources to review the manuscript.

We have revised the manuscript according to your comments and our responses in the interactive discussion. Please find a point-by-point reply to the referee comments and the editor below. The replies to the referees are essentially the same as in the interactive discussion.

Kind regards,
Maik Heistermann
(on behalf of the author team)

**Responses to the editor**

**EC:** *Your manuscript received positive feedback from the two experts and can be moved to the subsequent step of the journal. Please submit a revised version that incorporates the comments and suggestions for improvement received during the discussion phase. If needed, you can also improve other parts. For example, I suggest focusing more on the significance of your study. Both reviewers are willing to look at your revised version; therefore, another evaluation round will likely be activated.*

AR:  First, we would like to use the opportunity to thank the editor for guiding the review process of this manuscript, and for not giving up in the search for referees, which apparently was a tedious task.

We take from the editor's comment mainly the suggestion of "focusing more on the significance of your study". In our view, the most significant aspect of this study is that a local calibration might not always be the preferable strategy. This is line with referee #2 who stated that "the revelation that a local calibration may not be the best option in some circumstances is a key finding." While we think that the conclusions section sufficiently highlights this finding (and the complexity around it...), we agree that this could be emphasized more in the abstract, so we added a corresponding statement:

> Our results suggest that a local calibration – which currently is considered "best practice" – might quite often not be the preferable option.

Apart from this change, we used the opportunity to fix some typos and improve a few formulations in addition to the suggestions made by the referees.

Furthermore, the previous review file validation had suggested to check Fig. 1 with regard to its readability for readers with colour vision deficiencies. We checked, and the chosen color scheme worked mostly well, but showed some weakness for Protanopia (Red-blind) and Achromatopsia (Monochromacy). We hence adjusted the colors for Fig. 1. They now work well on all deficiencies, though Monochromacy remains a bit of a challenge.

**Responses to referee #1**

**RC:** *[...] Line 20: Typically we don't include dimensions for the a0, a1, and a2 as they are derived coefficients from Desilets 2010. The units make the equation balance, but I am not sure if they need to be included?*

**AR:** We also noticed that the units for $a_0$, $a_1$, and $a_2$ are mostly omitted in the literature. We do not have a strong opinion here, although omitting the dimensions would, as the referee implied, leave the equation unbalanced. Yet, for consistency with the existing literature, we dropped the units in the revised version of the manuscript.

**RC:** *Line 123, Eq 5.: McJannet and Desilets 2023 recently addressed a similar topic on using different neutron monitors and correcting for spatial variability. I am not sure exactly how these approaches are the same or differ, but the citation should be added with a short discussion of that paper. Also in the original COSMOS project an fscaling factor was included. This was also tied to the appropriate reference pressure selected using the COSMOS online calculator (http://cosmos.hwr.arizona.edu/Util/calculator.php). Use of the long-term average pressure for reference pressure instead of the one from the COSMOS online calculator did cause some problems when using the rover at different locations in my experience.*

**AR:** We appreciate this comment, and agree that McJannet and Desilets (2023) should be mentioned in this context. The best opportunity to do this is, in our view, in the conclusions section after line 333 of the preprint. There, we already stated:

> While the PARMA model is well-established, other such models exist, and future research might explore the potential sensitivity of the neutron intensity scaling on the choice of the model.

We would prefer, though, not to comprehensively discuss the methodological differences here, as a technical note should remain focused on the main idea. However, we'd like to emphasize that using the PARMA model is a subjective choice, and that future research should strive for a more systematic comparison of the existing approaches. This should include McJannet and Desilets (2023), but also e.g. Hawdon et al. (2014) and, as the referee mentioned, the scaling approach used in the original COSMOS paper by Zreda et al. (2012). The latter was based on the method that had been presented by Desilets and Zreda (2003) which is also the basis of the COSMOS online calculator (we mentioned this approach in ll. 40-41 of the preprint). Altogether, the above paragraph has become:

> We used the PARMA model (Sato et al., 2015) in order to account for the spatial variability of incoming neutron intensity (relative to a reference location), as a function of geographic latitude and longitude as well as terrain altitude. While the PARMA model is well-established, its application in this study remains a subjective and exemplary choice. Other similar models exist (e.g. Desilets and Zreda, 2003; Hawdon et al., 2014; McJannet and Desilets, 2023), and future research should aim to explore the potential sensitivity of the neutron intensity scaling to the choice of the model and the consistency of the resulting soil moisture estimates.

With regard to the choice of the reference pressure (last sentence of the above referee comment): The long-term average pressure could address possible effects of a biased barometer. If the effect of altitude is comprised in a static scaling factor (such as in our study), the factor $f_p$ that accounts for the *temporal* variability of barometric pressure should vary around a value of 1. However, if the barometer instrument is biased (which we found occasionally to be the case for various sensors), the long-term average will deviate from the standard atmospheric pressure, so $f_p$ will not vary around 1 and hence propagate the bias.

**RC:** *Line 140. The vegetation correction for biomass is still an active area of research so I would suggest a little more discussion here, particularly as its uncertainty is identified as being significant later in the manuscript. Hawdon et al. 2014 found a linear reduction in count rate at low biomass but it became more nonlinear for higher biomass and with sites with forest canopy. Franz 2015 found a linear reduction in count rate around 1% per kg/m$^2$ for croplands which is similar as reported by Baatz 2015. It seems there is a geometric effect for sites with a clumpy distribution of water, but I am not sure it is fully resolved yet (Franz et al 2013 provided some MCNPx simulations as well as Andreasen 2016, 2017). I am cautious about using the Baatz 2015 empirical equation for all vegetation types especially at high biomass sites within forests where this is likely a geometric factor reducing the impact of increasing biomass on the reduction in neutron counts (i.e. maybe a 0.3 or 0.5% in count rate per kg/m$^2$). I think so additional discussion of this effect some be added especially as it impacts the main conclusions of the paper.*

AR: Again, we appreciate this detailed comment and the provided references, and agree that the subject of vegetation correction should be addressed in more detail. The referee's comment provides an excellent basis for that, and we extented the paragraph around 140 ff. accordingly. At the same time, we would like to emphasize that one of the main conclusions of our study identifies the *quantification* of biomass (in high-biomass environments, i.e. forests) as a main source of uncertainty, and we suppose that this conclusion applies even if the reduction of neutron counts should level off to some degree for the presence of very high biomass levels in the sensor footprint. Furthermore, the relationship presented by Baatz et al. (2015) is based on a substantial number of observations with high levels of above-ground biomass (up to 30 kg/m²). Still, we fully agree with the referee that the effect of biomass and its horizontal and vertical distribution are not yet sufficiently understood, and that comprehensive neutron simulation studies might provide a more robust basis for corrections function that hold across various environments.

Altogether, we revised the paragraph at ll. 140 ff. as follows:

$f_b$ accounts for the effect of vegetation biomass on neutron count rates. The equation is based on the empirical analysis of a wide range of biomass levels by Baatz et al. (2015), according to which epithermal neutron count rates are reduced by 0.9 % for every kg of dry above-ground biomass per m² (AGB). This rate is similar to the reduction of 1 % per kg/m² reported by Franz et al. (2015) for croplands. It should be noted that, based on neutron transport modelling, Andreasen et al. (2017, 2020) found some effect of forest canopy structure on the reduction of epithermal neutron intensity. Apart from this effect of canopy structure, it also remains an open issue as to which extent simple linear reduction rates may apply for very high biomass levels.

**RC:** *L238. Avery 2016 also provided an uncertainty analysis and provided a CONUS map of soil properties for use with CRNS rovers and compared local sampling vs. available continuous datasets.*

AR: We thank the referee for pointing out this reference, which we added to the list of references which address the role of bulk density in the uncertainty of CRNS-based soil moisture estimation (in l. 238 of the preprint).

It should be kept in mind, though, that Avery et al. conceived the uncertainty of bulk density as the deviation between the bulk density taken from a global dataset (Global Soil Dataset for Earth System Modelling, Shangguan et al., 2014) and the bulk density obtained from in-situ samples. In our study, we do not define a specific source of uncertainty for bulk density, but we acknowledge that the footprint-wide average of bulk density as obtained from in-situ sampling could be quite uncertain itself, given the issue of horizontal and vertical representativeness of bulk density measurements.

**RC:** *In general: Crow 2012 describes the relationship between average soil moisture and its variance at different spatial aggregations. This relationship is asymmetric but parabolic shaped. Meaning that you would expect the highest standard deviation at intermediate soil moisture and low SD at low soil moisture and intermediate SD at high soil moistures. This information could be included in the expected range of SD across average soil moisture. I am not sure how this physical constraint would affect the local vs. general calibration suggestions. The same is also true for the bulk density and LW due to textural differences within a CRNS footprint. The point is there is both measurement error due to the instrument/method and natural variation due to spatial variability with a CRNS footprint.*

AR: We thank the referee for this comment. If we understand it correctly, the referee suggests to use scaling relationships such as the one shown by Crow et al. (2012) to guess (or constrain) the uncertainty of our estimated ground truth. However, we should be aware that the relationships between standard deviation and spatial mean, as shown in Fig. 3 of Crow et al. (2012), are empirical fits (not physical constraints, as the referee put it). Behind these seemingly smooth relationships is a lot of variability. This can be seen in the original publication by Famiglietti et al. (2008), e.g. in Fig. 6b where the standard deviation varies between 0.02 and 0.09 m³/m³ at the 800 m extent. In any case, we fully agree with the referee that "there is both measurement error due to the instrument/method and natural variation due to spatial variability with a CRNS footprint." We had already tried to emphasize this in the preprint (ll. 239-248) with regard to the uncertainty of $\theta_{cal}$:

> The error in the x-dimension relates to what we informally refer to as "ground truth", although the actual level of truth in $\theta_{\mathrm{cal}}$ remains difficult to determine. All we know is that numerous errors might accumulate along the way, e.g. the measurement error of $\theta$ at a single point (possibly systematic, depending on technology), the effects of limited sample size in combination with the limited horizontal and vertical representativeness of each measurement, or the uncertainty of the horizontal and vertical weighting functions.

Overall, we think that an in-depth discussion of the scaling behaviour of soil moisture uncertainty is beyond the scope of this technical note. When we choose between local and global calibration, the magnitude of $\sigma_{\theta_{\mathrm{cal}}}$ certainly is an important aspect, yet any guess at it should consider the specific local conditions.

At the same time, we much appreciate the referee's idea to make use of the aforementioned relationships. In fact, if we acknowledge that (i) spatial variability peaks at intermediate soil moisture conditions, and (ii) that the uncertainty of $\theta_{\mathrm{cal}}$ propagates less with increasing soil wetness (one of our conclusions), it appears that a local calibration is, if at all, most recommendable under wet soil conditions. In the revised manuscript, we added a corresponding sentence after l. 347 of the preprint:

> [...] if the calibration were carried out under dry conditions, the error would grow substantially under wet application conditions. Considering further that the spatial variability of soil moisture appears to reach a maximum under intermediate wetness conditions (see e.g. Crow at al., 2012; Famiglietti et al., 2008), it could be recommendable to obtain $\theta_{\mathrm{cal}}$ under rather wet conditions [...]

**Responses to referee #2**

**RC:**  *[...] Line 60: Sentence is a bit clunky - suggest reword to "This requires the user to account for the relative sensitivity of the neutron detector, the effects of other hydrogen pools in the sensor footprint, and the effects of geographic latitude, longitude, and altitude"*

 **AR:**  The suggestion was implemented.

**RC:**  *Line 130: An alternative method for correcting for spatial and altitudinal variability has been published by McJannet and Desilets (2023) and it may be useful here as it will allow users to apply even more widely to areas outside of Europe.*

 **AR:**  This suggestion is in line with a comment made by referee #1, and we agree that McJannet and Desilets (2023) should be mentioned in the context of this study. The best opportunity to do this is, in our view, in the conclusions section after line 333 of the preprint. There, we already stated:

> While the PARMA model is well-established, other such models exist, and future research might explore the potential sensitivity of the neutron intensity scaling on the choice of the model.

In the context of this paragraph, we now specifically refer to alternative approaches, including McJannet and Desilets (2023), but also e.g. Hawdon et al. (2014) and the scaling approach used in the original COSMOS paper by Zreda et al. (2012). The latter was based on the method that had been presented by Desilets and Zreda (2003). Altogether, the above paragraph has become:

> We used the PARMA model (Sato et al., 2015) in order to account for the spatial variability of incoming neutron intensity (relative to a reference location), as a function of geographic latitude and longitude as well as terrain altitude. While the PARMA model is well-established, its application in this study remains a subjective and exemplary choice. Other similar models exist (e.g. Desilets and Zreda, 2003; Hawdon et al., 2014; McJannet and Desilets, 2023), and future research should aim to explore the potential sensitivity of the neutron intensity scaling to the choice of the model and the consistency of the resulting soil moisture estimates.

**RC:** *Line 145 suggest changing to "For our reference detector we chose a so called..."*

AR: The suggestion was implemented.

**RC:** *Line 334: suggest "Altogether, we assume that we have considered the most relevant processes and variables..."*

AR: The suggestion was implemented.

**RC:** *Line 335: suggest "On average, the general calibration function fits the weighted average of locally measured SWC per CRNS footprint fairly well, specifically for the apparent gravimetric SWC".*

AR: The suggestion was implemented.

**RC:** *Line 340: suggest "Interestingly, though, we only become aware of these errors in the case where we apply a general calibration..."*

AR: The suggestion was implemented.

**RC:** *Line 344: suggest "Based on the results of this study, we recommend considering both calibration options, local and general, and weighing the relative uncertainty of the one against the other"*

AR: The suggestion was implemented.

**References**

Andreasen, M., Jensen K. H., Desilets D., Zreda M., Bogena H. R., Looms M. C. (2017): Cosmic-ray neutron transport at a forest field site: the sensitivity to various environmental conditions with focus on biomass and canopy interception. Hydrology and Earth System Sciences, 21(4):1875–1894.

Andreasen, M., Jensen, K. H., Bogena, H., Desilets, D., Zreda, M., and Looms, M. C. (2020): Cosmic ray neutron soil moisture estimation using physically based site-specific conversion functions. Water Resources Research, 56, e2019WR026588. https://doi.org/10.1029/2019WR026588

Avery, W. A., Finkenbiner C., Franz T. E., Wang T., Nguy-Robertson A. L., Suyker A., Arkebauer T., Muñoz-Arriola F. (2016): Incorporation of globally available datasets into the roving cosmic-ray neutron probe method for estimating field-scale soil water content. Hydrology and Earth System Sciences, 20(9):3859–3872. https://doi.org/10.5194/hess-20-3859-2016

Crow, W. T., Berg, A. A., Cosh, M. H., Loew, A., Mohanty, B. P., Panciera, R., Rosnay, P. de, Ryu, D., Walker, J. P. (2012): Upscaling Sparse Ground-Based Soil Moisture Observations For The Validation Of Coarse-Resolution Satellite Soil Moisture Products. Reviews of Geophysics, 50, https://doi.org/10.1029/2011rg000372

Desilets, D. and Zreda, M. (2003): Spatial and temporal distribution of secondary cosmic-ray nucleon intensities and applications to in-situ cosmogenic dating, Earth Planet. Sc. Lett., 206, 21–42.

Famiglietti, J. S., Ryu, D., Berg, A. A., Rodell, M., and Jackson, T. J. (2008): Field observations of soil moisture variability across scales, Water Resour. Res., 44, W01423, doi:10.1029/2006WR005804.

Franz T. E., Zreda M., Rosolem R., Hornbuckle B. K. , Irvin S. L., Adams H., Kolb T. E., Zweck C., Shuttleworth W. J. (2013): Ecosystem-scale measurements of biomass water using cosmic ray neutrons. Geophysical Research Letters, 40(15), https://doi.org/10.1002/grl.50791

Franz, T. E., Wang, T., Avery, W., Finkenbiner, C., Brocca, L. (2015): Combined analysis of soil moisture measurements from roving and fixed cosmic ray neutron probes for multiscale real-time monitoring. Geophysical Research Letters, 42, https://doi.org/10.1002/2015GL063963

Hawdon, A., McJannet D., Wallace J. (2014) Calibration and correction procedures for cosmic-ray neutron soil moisture probes located across Australia. Water Resources Research, 50(6):5029–5043, https://doi.org/10.1002/2013wr015138

McJannet, D. L., Desilets D. (2023): Incoming Neutron Flux Corrections for Cosmic-Ray Soil and Snow Sensors Using the Global Neutron Monitor Network. Water Resources Research, 59(4):e2022WR033889, https://doi.org/10.1029/2022WR033889.

Shangguan, W., Dai, Y., Duan, Q., Liu, B. and Yuan, H. (2014): A Global Soil Data Set for Earth System Modeling. Journal of Advances in Modeling Earth Systems, 6: 249-263.

Zreda, M., Shuttleworth, W. J., Zeng, X., Zweck, C., Desilets, D., Franz, T., and Rosolem, R. (2012): COSMOS: the COsmic-ray Soil Moisture Observing System, Hydrol. Earth Syst. Sci., 16, 4079–4099, https://doi.org/10.5194/hess-16-4079-2012.